# The Only Chemoreceptor Encoded by *che* Operon Affects the Chemotactic Response of *Agrobacterium* to Various Chemoeffectors

**DOI:** 10.3390/microorganisms9091923

**Published:** 2021-09-10

**Authors:** Jingyang Ye, Miaomiao Gao, Qingxuan Zhou, Hao Wang, Nan Xu, Minliang Guo

**Affiliations:** College of Bioscience and Biotechnology, Yangzhou University, Yangzhou 225009, China; DX120170118@yzu.edu.cn (J.Y.); miaomiaogao1@163.com (M.G.); m18362821722@163.com (Q.Z.); wanghao@yzu.edu.cn (H.W.); nanxu@yzu.edu.cn (N.X.)

**Keywords:** *Agrobacterium fabrum*, chemotaxis, chemoreceptor, cellular localization, methyl-accepting chemotaxis protein, chemoeffector

## Abstract

Chemoreceptor (also called methyl-accepting chemotaxis protein, MCP) is the leading signal protein in the chemotaxis signaling pathway. MCP senses and binds chemoeffectors, specifically, and transmits the sensed signal to downstream proteins of the chemotaxis signaling system. The genome of *Agrobacterium fabrum* (previously, *tumefaciens*) C58 predicts that a total of 20 genes can encode MCP, but only the MCP-encoding gene *atu0514* is located inside the *che* operon. Hence, the identification of the exact function of *atu0514*-encoding chemoreceptor (here, named as MCP_514_) will be very important for us to understand more deeply the chemotaxis signal transduction mechanism of *A. fabrum*. The deletion of *atu0514* significantly decreased the chemotactic migration of *A. fabrum* in a swim plate. The test of *atu0514*-deletion mutant (Δ514) chemotaxis toward single chemicals showed that the deficiency of MCP_514_ significantly weakened the chemotactic response of *A. fabrum* to four various chemicals, sucrose, valine, citric acid and acetosyringone (AS), but did not completely abolish the chemotactic response. MCP_514_ was localized at cell poles although it lacks a transmembrane (TM) region and is predicted to be a cytoplasmic chemoreceptor. The replacement of residue Phe328 showed that the helical structure in the hairpin subdomain of MCP_514_ is a direct determinant for the cellular localization of MCP_514_. Single respective replacements of key residues indicated that residues Asn336 and Val353 play a key role in maintaining the chemotactic function of MCP_514_.

## 1. Introduction

Chemotaxis is an adaptive behavior of motile bacteria moving along the concentration gradient of chemoeffectors towards an optimal environment [1,2]. This adaptive behavior is regulated by a two-component system composed of histidine kinase CheA and chemotaxis response regulator protein CheY [3,4]. When a chemical attractant exists in the environment, methyl-accepting chemotaxis protein (MCP) can bind the attractant, and the ligand binding will change the conformation of MCP; then, through a coupling protein CheW, the kinase activity of CheA will be suppressed. The suppression of CheA activity will delay the transfer of the phosphoryl group to the response regulator protein CheY, resulting in the decrease in phosphorylated CheY [5,6]. The absence of phosphorylated CheY makes less change of the rotational direction of flagella and thus keeps bacteria swimming to the attractant. On the contrary, when the concentration of attractant goes down (or repellent goes up), MCP activates the kinase activity of CheA. Self-phosphorylated CheA passes the phosphoryl groups to CheY. Phosphorylated CheY will bind to the flagellar motor and change the rotational direction of flagella frequently, which causes cell tumbling and changes direction away from the adverse environment [7,8].

Many bacteria possess chemotactic behavior for a large number of chemicals, such as different nitrogen or carbon nutrient substances, different environmental pollutants, or different signal chemicals released by their ecological partners [9,10,11,12,13]. To sense the vast chemoeffectors, bacteria must evolve an enormous number of different chemoreceptors. A typical chemoreceptor contains an N-terminal periplasmic ligand binding domain (LBD), a transmembrane (TM) helical region, a HAMP (existing in histidine kinases, adenylate cyclases, methyl-accepting chemotaxis proteins, and phosphatases) domain, and a C-terminal cytoplasmic signaling domain (SD), comprising a methyl-accepting (MA) subdomain, a flexible bundle (FB) subdomain, and a hairpin (HP) subdomain (Figure 1A) [7,14,15]. The hairpin subdomain interacts with the coupling protein CheW and the senor kinase CheA. Chemoreceptors use LBD to recognize chemoeffectors. In order to recognize different chemicals, chemoreceptors have evolved many different types of LBDs [9]. According to the LBD and membrane topology, chemoreceptors can be divided into four classes, of which class IV is cytoplasmic (soluble) chemoreceptors and the rest are transmembrane chemoreceptors. Based on the length from the N-terminus to the MA domain, cytoplasmic chemoreceptors can be divided into two subclasses of IVa and IVb. The N-terminal domain of IVa has at least 108 amino acids, while the N-terminal domain of IVb is shorter than 108 amino acids [16,17]. Among 8384 chemoreceptors defined by the MA subdomain in the complete genome of the SMART database [18], 14% of the chemoreceptors are cytoplasmic chemoreceptors. The ligand-binding motif is generally predicted at the N-terminal of IVa subclass chemoreceptors. Based on the statistical analysis of 1129 chemoreceptors with LBD in the SMART database, 47% of LBD is PAS domain, such as AerC in *A. brasilense* [19]; CZB domain found in *Helicobacter pylori* TlpD [20], is the second most common LBD, accounting for 8% of all IVa chemoreceptors; protoglobin [21] is the third most common LBD, accounting for 7% of all IVa chemoreceptors; the remaining domains account for less than 1% [22].

*Agrobacterium fabrum* is a Gram-negative bacterium and induces crown gall tumor disease in most dicotyledonous plants by genetically transforming the host [23,24]. In the natural environment, *A. fabrum* is usually distributed around the rhizosphere of a plant, and chemotaxis is an important first process of its interaction with the host. It has two kinds of lifestyles; one is to survive in the soil environment independently and the other is symbiotic with the plant as a pathogen [25,26,27]. Only when it recognizes and senses the sugars, acidic pH or phenols released by the host can *A. fabrum* begin the infecting process by chemotaxis toward the wound site of the host [28]. In the late 1980s, chemotaxis of *A. fabrum* was preliminarily studied, proving that it can respond to sugars, amino acids and phenols released by the injured plant tissues [29,30,31], but the specific chemotaxis mechanism has been rarely studied [32]. The genome sequence of *A. fabrum* C58 predicts that it has only one chemotaxis gene cluster (*che* cluster). The gene organization of this gene cluster is shown in Figure 1B. This gene cluster contains the genes encoding most components of the chemotaxis system. Spaces between the adjacent genes are very short and some adjacent genes even share a few nucleotides. All the genes in this cluster are predicted to be controlled by the same upstream promoter and thus located on the same operon [12,33]. In addition to the *atu0514* gene, the only gene annotated to encode MCP in this *che* operon, *A. fabrum* carries an additional 19 MCP-encoding genes, including 1 on the Ti plasmid, 1 on the At plasmid, 5 on the linear chromosome, and the remaining 12 on the circular chromosome [33]. The number of MCPs in *A. fabrum* C58 indicates its complex chemotaxis signal and strong environmental adaptability [34].

The *A. fabrum atu0514* gene, located on the circular chromosome, is the only MCP-encoding gene in the *che* operon [12]. Unlike all other MCPs, MCP_514_ is co-expressed with all other core chemotaxis components. Therefore, it may be the most important chemoreceptor, and characterizing its function is very helpful for us to further understand the chemotaxis signal transduction mechanism of *A. fabrum*. In this study, we firstly constructed the *atu0514*-deletion mutant Δ514 and the complemented strain Δ514-C and tested the effect of *atu0514*-deficiency on the chemotactic response of *A. fabrum* C58. We also identified the key residues that affect the cellular localization and function of MCP_514_.

## 2. Materials and Methods

### 2.1. Primers, Plasmids, Bacterial Strains and Growth Conditions

The primers, plasmids and bacterial strains used in this study are listed in Appendix A. Lysogeny broth (LB) liquid or agar medium was used to grow *E. coli* at 37 °C [35]. *A. fabrum* was grown in MG/L or AB-sucrose liquid or agar medium at 28 °C [36,37]. Concentrations of ampicillin and kanamycin used for *E. coli* were 100 and 50 µg/mL, respectively. Concentrations of kanamycin and carbenicillin for *A. fabrum* were 100 µg/mL.

### 2.2. DNA Manipulations

DNA manipulations followed the standard molecular protocols [35]. Plasmid isolation was performed with the TIANprep Mini Plasmids Kit (TIANGEN BIOTECH Corporation, Beijing, China). PCR products obtained by Veriti 96-well cycler (Thermo Fisher Scientific Inc., Waltham, MA USA) and DNA fragments were purified from agarose gels by using the TaKaRa MiniBEST Agarose Gel DNA Extraction Kit (TaKaRa Corporation, Dalian, China). Plasmids were transferred into *E. coli* competent cells by heat-shock [35] and into *A. fabrum* by the Eppendorf electroporation instrument Eporator^®^ (Eppendorf AG, Hamburg, Germany) [36].

### 2.3. Mutagenesis and Complementation of Atu0514

Based on the principle of homologous recombination, we used the pEX18Km-derived gene replacement plasmids to construct the corresponding gene deletion mutants [38,39]. Plasmid pEX18Km carries both a positive selection marker (kanamycin resistance, K_m_^R^) and counterselectable marker (suicide gene *sac*B) and cannot be replicated in *A. fabrum*. The positive selection marker allowed to select the transformants, in which the whole plasmid was integrated into the genome by the first homologous recombination. The counterselectable marker allowed to counterselect the transformants, in which both the target DNA fragment and the undesirable plasmid backbone were deleted from the genome by the second homologous recombination. The combined utilization of selectable and counterselectable markers can precisely generate an unmarked mutant without any undesirable DNA fragment (Appendix A). *A. fabrum* wild type C58 was used to construct the *atu0514*-deletion mutant ∆514. *A. fabrum cheW_1_*-*cheW_2_* double-deletion mutant Δw [40] was used to construct *atu0514*-*cheW_1_*-*cheW_2_* triple-deletion mutant Δ514Δw. The desirable mutant was screened using PCR (Appendix A) and verified by sequencing. The DNA fragment encoding amino acids 14–453 of MCP_514_ was precisely deleted in both ∆514 and Δ514Δw mutants (Appendix A). The complementation of MCP_514_ in the *atu0514*-deletion mutant was fulfilled by the introduction of plasmid expressing MCP_514_ (or its variants). Gene fusions with the *egfp* as well as deletion constructs were created by overlap extension PCR, as described by Higuchi [41]. These *egfp*-fused genes were cloned into the modified vector pUCA19 to generate the plasmids expressing eGFP-fused proteins. When these eGFP-fused proteins were expressed in *A. fabrum* cells, the cellular localizations of these eGFP-fused proteins could be observed by using Zeiss confocal microscope LSM 880 NLO (Carl Zeiss AG, Oberkochen, Germany).

### 2.4. Chemotaxis Assays

The procedure of capillary assay was followed as described by Adler in 1973 [42], with minor modifications. *A. fabrum* cells were harvested from mid-log-phase culture by centrifugation at 4000 rpm for 3 min at room temperature (25 °C) and suspended in chemotaxis buffer (0.1 mmol/L EDTA, 10 mmol/L KH_2_PO_4_, pH 7.0) to an OD_600nm_ of 0.1. *A. fabrum* cell suspension (300 μL) was used to make a bacterial pond. The capillary tube was sealed at one end and filled with attractant at peak concentration dissolved in the chemotaxis buffer. The open end of the capillary tube was inserted into the bacterial pond and incubated for 1 h at room temperature (25 °C), then, the solution in the capillary tube was expelled and completely transferred into 1 mL of AB-sucrose medium. Dilutions were plated in duplicate on MG/L plates and incubated for 2 days at 28 °C. The colonies in the plates were counted and represented the number of cells attracted to the capillary tube.

The procedure of swim agar plate assay was followed that described by Merritt [43], with minor modifications. The tested strains were inoculated in AB-sucrose liquid medium and grown to the middle log phase, and then, the OD_600nm_ was adjusted to 0.6 by using AB-sucrose liquid medium. A total of 3 μL of bacterial suspension was dropped onto an AB-sucrose swim plate containing 0.2% agar, and 5 replicates were set. After incubation at 28 °C for 36–48 h, the diameter of the bacterial colony circle was measured, and the data were statistically analyzed.

### 2.5. Bacterial Two-Hybrid Analyses

The bacterial two-hybrid system from the Stratagene^®^ (Agilent Technologies Inc., Santa Clara, CA, USA) was used for testing protein–protein interactions. All the operations were conducted according to the manual. The open read frame (ORF) of *atu0514* was inserted into the bait plasmid pBT to express λcI-MCP_514_ (bait) fusion protein. ORFs of *cheW_1_* and *cheW_2_* were inserted to the target plasmid pTRG to express CheW_1_ (target)–RNAP and CheW_2_ (target)–RNAP fusion proteins, respectively. The interaction between bait (MCP_514_) and target (CheW_1_ or CheW_2_) will take λcI and RNAP together to induce the expression of β-galactosidase, and thus, the bacterial colonies growing on plates containing 80 μg/mL X-gal will be blue. Otherwise, the bacterial colonies will be of normal color. Galactosidase activity was determined by the method of Miller [44].

### 2.6. Fluorescence Microscopy

For microscopy observation, agrobacterial cells from the mid-log-phase cultures were added to the center of the slides. A coverslip was placed on top of the culture droplet. The edges of the coverslip were sealed with acrylic polymer to prevent drying. *A. fabrum* cells were visualized by a Zeiss LSM 880 NLO system (Carl Zeiss AG, Oberkochen, Germany) using an Ar laser (excitation wavelength of 488 nm and emission wavelength of 500 to 550 nm) and a ×100 oil immersion objective. The images were analyzed and edited using ZEN lite (Blue edition), version 3.2 (Carl Zeiss AG, Oberkochen, Germany).

### 2.7. Statistical Analysis

The quantitative data shown in this study were the means with the standard deviations (SDs), which were derived from at least three independent experiments conducted in triplicate. Differences among bacterial strains were compared using one-way analysis of variance (ANOVA), followed by the Tukey test for multiple comparisons. The statistical analysis was conducted using Microsoft Office Excel’s data analysis tool (2019 version) (Microsoft Corporation, Redmond, WA, USA).

## 3. Results

### 3.1. MCP_514_ Is a Cytoplasmic Chemoreceptor, but Localized at Cell Poles

By SMART analysis, MCP_514_ has a total of 514 amino acids and carries three conserved domains, protoglobin domain, HAMP domain and cytoplasmic signal domain (SD), in the order from N-terminal to C-terminal (Figure 1C). The protoglobin domain of MCP_514_ shares 13.51% sequence identity with the LBD of HEMAT, a cytoplasmic chemoreceptor from *B. subtilis* [21], through the SWISS-Model homology search. According to Alexandre’s heptapeptide classification, cytoplasmic signal domain (SD) belongs to the 36H family [45]. The two best-studied MCPs of the 36H family are Tsr and Tar proteins of *E. coli*. Further analysis on the secondary structure of MCP_514_ by SOSUI [46] and SPLIT [47] shows that MCP_514_ does not contain the hydrophobic domain, indicating that MCP_514_ does not possess a transmembrane (TM) region and belongs to an IVa cytoplasmic chemoreceptor.

Transmembrane chemoreceptors are mainly localized at cell poles, but the cytoplasmic chemoreceptors have a wider cellular localization mode, ranging from co-localization with transmembrane chemoreceptor arrays to a diffuse cytoplasmic distribution [48]. The localization of some cytoplasmic chemoreceptors is associated with the physiology and life cycle of bacteria [22]. According to previous classification [9], MCP_514_ should be classified into the category of the cytoplasmic chemoreceptor due to the lack of a transmembrane region, but we do not know the cellular localization of MCP_514_. To observe the cellular localization of MCP_514_, the enhanced green fluorescent protein (eGFP) was fused to the N-terminus of MCP_514_ because only the N-terminally GFP-tagged MCP_514_ was functional [49]. This eGFP-MCP_514_ fusion protein was expressed in the MCP_514_-deficient strain ∆514. Figure 2A showed that the eGFP-MCP_514_ fusion protein is localized at the poles of the *A. fabrum* cell. However, the eGFP-MCP_514_ fusion protein is distributed in the whole cell of *E. coli* (Figure 2B), verifying that MCP_514_ lacks a transmembrane domain. These data also imply that the polar localization of MCP_514_ in *A. fabrum* cell requires the assistance of other *A. fabrum* proteins.

### 3.2. MCP_514_ Significantly Affects the Chemotactic Response of Agrobacterium Fabrum

Since MCP_514_ is localized at cell poles, we next try to test the effect of MCP_514_ on the chemotactic response of *A. fabrum*. When bacterium grows on the swim agar plate, the utilization of nutrient substances by bacterium will result in a nutrient concentration gradient. Bacterium with chemotaxis will move outward along the nutrient concentration gradient and grow a big colony. Consequently, the overall chemotactic response to nutrient substances can be characterized by measuring the colony size in the swim agar plate [10]. *A. fabrum* has only one CheA, and CheA is the key component of a chemotaxis system. The deletion of the *cheA* gene will completely abolish the chemotactic response of *A. fabrum,* and thus, the CheA-deficient mutant Δ*a* was used as a control of chemotaxis deficiency [40]. As shown in Figure 3, the deficiency of MCP_514_ significantly attenuates the overall chemotactic response of *A. fabrum* to nutrient substances. The complementation of MCP_514_ by the introduction of MCP_514_-expressing plasmid can fully restore the chemotactic response of the MCP_514_-deficient mutant to the level of the wildtype, confirming the role of MCP_514_ in the chemotactic response. These results also provided evidence that the deletion of *atu0514* did not have a polar effect on the rest of the *che* operon. Previous research showed that *A. fabrum* has chemotaxis toward sugars, amino acids, organic acids and phenols [29,30]. The traditional capillary assay is an effective method for quantifying the chemotaxis ability of bacteria [42]. The chemotactic responses of wildtype C58, mutant Δ514 and complemented strain Δ514-C to four various chemicals (1 µmol/L sucrose, 1 mmol/L valine, 1 mmol/L citric acid and 0.1 µmol/L acetosyringone (AS)) were measured by using the traditional capillary method. As shown in Figure 4, the deletion of *atu0514* significantly weakens the chemotaxis of *A. fabrum* toward these four different types of chemicals. However, the chemotaxis of strain Δ514 toward these chemicals does not completely disappear, indicating that MCP_514_ is not the receptor directly recognizing these substances but affects the chemotaxis efficiency in other ways.

### 3.3. Both CheW_1_ and CheW_2_ Interact with MCP_514_ but Do Not Affect the Cellular Localization of MCP_514_

It is known that CheW couples CheA to chemoreceptors and forms stable ternary signaling complexes with chemoreceptors and CheA [3]. To confirm the role of MCP_514_ in the chemotaxis signal transduction pathway of *A. fabrum*, the bacterial two-hybrid system was used to test the interaction between MCP_514_ and two CheWs. As shown in Figure 5A, the colors of the colonies representing MCP_514_/CheW_1_ and MCP_514_/CheW_2_ interaction are bluer than that of the negative control. The quantification of β-galactosidase activity also showed that the β-galactosidase activities of the colonies expressing these two tested protein pairs (MCP_514_/CheW_1_ and MCP_514_/CheW_2_) are significantly higher than that of the negative control (Figure 5B). This indicates that MCP_514_ protein interacted with both CheW_1_ and CheW_2_ proteins.

Both CheW_1_ and CheW_2_ interact with MCP_514_ and most of the ternary MCP-CheW-CheA complexes are localized in the cell poles. However, MCP_514_ lacks a transmembrane domain. It is unknown whether the cellular localization of MCP_514_ is dependent on the ternary MCP–CheW–CheA complex. Therefore, we tested the effects of the CheW deficiency on the cellular localization of MCP_514_. To determine whether CheW affects the cellular localization of MCP_514_, a plasmid expressing eGFP–MCP_514_ fusion protein was transferred into the *atu0514–cheW_1_–cheW_2_* triple deletion mutant Δ514Δ*w*. Fluorescence observation showed that MCP_514_ in the CheW-deficient strain is still localized at cell poles, indicating that CheW deficiency does not affect the polar localization of MCP_514_ (Figure 6).

### 3.4. Helical Structure of Hairpin Subdomain Is Required for the Cellular Localization of MCP_514_

Due to the lack of a transmembrane domain in MCP_514_ and the evidence that CheW deficiency does not affect the polar localization of MCP_514_, it is most likely that MCP_514_ is localized in the cell poles via interacting with other MCPs. Results from *E. coli* MCPs showed that the hairpin subdomain of MCP is a coiled-coil of two antiparallel helices with a ‘U-turn’ and two hairpin subdomains form a supercoiled four-helical bundle, which makes MCP form homodimeric molecules [50]. The dimers of different MCPs can form mixed trimers of dimers via the interactions between their highly conserved helical bundle, and several residues play key roles in the formation of trimer (Appendix A) [51,52,53].

To determine if the hairpin subdomain of MCP_514_ affects the cellular localization of MCP_514_, key residue Phe328 in the hairpin subdomain was changed to Ala, Pro or Trp. These three single-residue-substituted MCP_514_ variants were fused to eGFP, respectively. Fluorescence observation showed that the substitution of Phe328 for Pro causes the MCP_514_ diffusion in the cytoplasm (Figure 7C), whereas the replacement of Phe328 by Ala or Trp does not affect the polar location of MCP_514_ (Figure 7A,B). Proline is a constraint on the formation of helix. Replacement of Phe328 by Pro will destroy the helical structure of the hairpin subdomain. These results demonstrated that the helical structure of the hairpin subdomain is required for the cellular localization of MCP_514_.

### 3.5. Two Key Residues of Hairpin Subdomain Play a Key Role in Maintaining the Chemotactic Function of MCP_514_

The cellular localization of MCP_514_ is dependent on the hairpin subdomain, which is involved in the interactions between different MCPs, as well as the interactions with the CheA and CheW [54]. The chemotactic signal is collaboratively transduced by the mixed MCP teams, and all signals from different MCPs will converge to CheA [51]. It is reasonable that residues involving in the interaction between MCPs may affect the chemotactic function of MCP_514_.

To further determine the key residues of MCP_514_ that are involved in the trimer contact, we aligned the sequence of MCP_514_ with the sequences of Tsr and Tar from *E. coli* [55] and chose Phe328, Asn336, Glu340, Arg343 and Val353 of MCP_514_ as the target residues of the site-directed mutation (Appendix A). Five residues were respectively replaced by alanine to generate five single residue-substituted MCP_514_ variants. These MCP_514_ variants were expressed in the MCP_514_-deficient mutant (Δ514) by the introduction of the MCP_514_ variant-expressing plasmid, respectively. Colonies of the MCP_514_-deficient mutant expressing various MCP_514_ variants are shown in Figure 8A. The diameters of these tested strain colonies were used to quantify the effects of various MCP_514_ variants on the chemotactic response of *A. fabrum* [43]. As shown in Figure 8, two single residue-substituted MCP_514_ variants, MCP_514_^N336A^ and MCP_514_^V353A^, are unable to restore the chemotactic response of the MCP_514_-deficient mutant to the level of the wildtype, demonstrating that residues Asn336 and Val353 play a key role in maintaining the chemotactic function of MCP_514_.

## 4. Discussion

In natural environments, *A. fabrum* is usually distributed around the rhizosphere of a plant [24]. Chemotaxis is the important initial step for *A. fabrum* to infect the plant host [56]. Only when *A. fabrum* correctly recognizes and responds to the chemical signals released by the plant host can it contact the host plant, infect the host and start the tumorigenic processes [27,57,58,59]. MCPs are the first components of the chemotaxis system. The recognition of chemicals by these proteins is the initial stage of chemotaxis signaling transduction. Adaptive modification of the conserved glutamate domain of the MCP signal domain ensures that they are highly sensitive to different concentrations of chemoeffectors [60]. Although *A. fabrum* C58 contains 20 MCP-encoding genes, only one MCP-encoding gene (*atu0514*) is in the *che* operon [12]. It is rational that the MCP_514_ encoded by the *atu0514* gene may play a unique role in the signaling transduction of chemotaxis.

Our results show that MCP_514_ deficiency significantly affects not only the overall chemotactic response of A. fabrum to nutrient substances (Figure 3) but also the chemotaxis toward four various types of chemicals (Figure 4). The ligand binding domain (LBD) of MCP_514_ is predicted to be a protoglobin, which cannot bind these four tested types of chemicals (Figure 1B). MCPs with different LBDs can form mixed trimers of dimers, and thousands of MCPs form a receptor cluster. The receptor cluster comprised of MCPs with different detection specificities collaboratively transduces the chemotactic signal in a team signaling model [51]. The effects of MCP_514_ deficiency on the chemotaxis of *A. fabrum* toward four various types of chemicals demonstrate that MCP_514_ is a very important member of the chemoreceptor signaling team, and the absence of MCP_514_ will affect the signaling efficiency of the whole chemoreceptor signaling team, explicating the reason of the *atu0514* gene locating the *che* operon.

Based on the sequenced genomes, 43% of archaeal and 14% of bacterial MCPs lack a transmembrane (TM) region. These TM-lacking MCPs are classified as cytoplasmic chemoreceptors. Unlike the transmembrane chemoreceptors, which are mainly located at the pole of the cell, cytoplasmic MCPs adopt more exotic locations [22]. Some cytoplasmic chemoreceptors are polarized at one end of the cell, such as, the HemAt of *B. subtilis* and the IcpA of *S. meliloti* [21,61]; other cytoplasmic chemoreceptors have both polar and diffuse states, for example, the AerC of *A. brasilense* was diffused in cytoplasm under an oxygen-rich environment but located in the cell polar under symbiotic nitrogen-fixing state, which helps cells adapt to hypoxic environments [19]. There are also some bacteria whose cytoplasmic chemoreceptors formed clusters in the cytoplasm, for example, the TlpC and TlpT of *R. sphaeroides* are located in the center of the cell by forming cytoplasmic clusters [62]. It is believed that the subcellular localization and distribution of the cytoplasmic chemoreceptor arrays are associated with the life cycle of bacteria [22]. Our results show that MCP_514_ localizes cell poles. Although both CheW_1_ and CheW_2_ interact with MCP_514_, respectively, (Figure 5), the cellular localization of MCP_514_ is independent of CheW (Figure 6), which is consistent with the previous results obtained from *E. coli*, in which trimers of homodimers of MCPs form clusters of MCPs and in turn recruit CheA/CheW to assemble MCP-CheW-CheA ternary complexes [52,54]. In combination with the previous studies on other cytoplasmic MCPs [22,62], our results could support that this TM-lacking chemoreceptor, MCP_514_, localizes the cell poles through interacting with other transmembrane chemoreceptors, although the experimental evidence may be required to demonstrate the interaction of MCP_514_ with other MCPs.

According to the previous studies on Tsr and Tar of *E. coli*, the hairpin (HP) subdomain of MCP is not only the region forming homodimeric MCP but also the contacting sites for the formation of mixed trimers as well as the assembly of MCP-CheW-CheA ternary complexes [51,55]. Five conserved amino acid residues in the two antiparallel helices of *E. coli* Tsr (Phe373, Asn381, Glu385, Arg388 and Val353) are important for the cluster formation and function of MCP [51]. Hydrophobic interactions between the helices contribute to the main trimer packing forces [55]. Five corresponding residues of MCP_514_ conducted the single residue of the respective substitution. Phe328 of MCP_514_ is in the key site of the helical structure of the HP subdomain. The replacement of Phe328 by the helix destroyer, proline, results in MCP_514_ diffusing in the cytoplasm. Replacement of Phe328 by alanine does not affect both the cellular location and chemotactic function of MCP_514_. These results demonstrate that Phe328 is not the direct determinant to the chemotactic function of MCP_514_, but indirectly affects MCP_514_ function through stabilizing the helical structure. Amongst the five tested MCP_514_ variants, three variants have the full function of the wildtype, which is slightly different from the results of *E. coli* Tsr [51].

## 5. Conclusions

MCP_514_ is localized at cell poles, although it lacks a transmembrane region. The cellular localization of MCP_514_ is independent of the assembly of MCP-CheW-CheA ternary complex but dependent on the interaction with other MCPs through the hairpin (HP) subdomain. Compared with other MCPs, MCP_514_ plays a superior role in maintaining the signaling efficiency of the whole chemoreceptor signaling team. The helical structure of the hairpin subdomain is the prerequisite of MCP_514_ cellular localization and functioning. The molecular mechanism of MCP_514_ interacting with other MCPs and functioning in the chemotaxis signaling is similar to that of *E. coli* MCPs, although the roles of some key residues of MCP_514_ in the transducing signal are slightly different from those of *E. coli* Tsr.

## Figures and Tables

**Figure 1 microorganisms-09-01923-f001:**
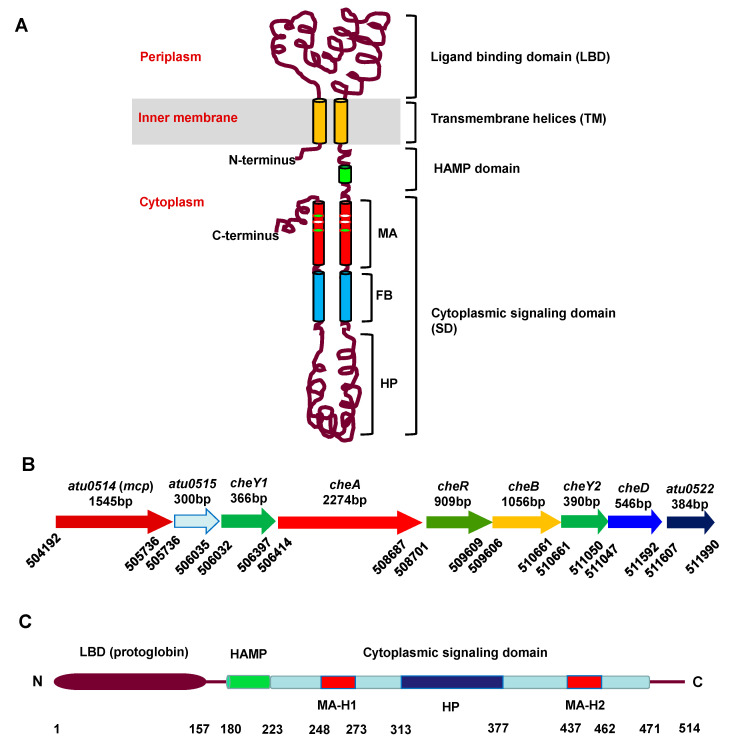
Topological structure of a typical MCP, gene organization of a *A. fabrum che* gene cluster and domain organization of MCP_514_. (**A**) Architectural model of a typical MCP. Cylinders represent α-helices. MA: methyl-accepting (or sensory adaptation) subdomain; FB: flexible bundle subdomain; HP: hairpin subdomain. (**B**) Gene organization of *A. fabrum che* gene cluster. Arrows represent genes. Above the arrows are the names and lengths of the genes. Numbers under the arrows indicate the nucleotide position of the gene border. (**C**) Predicted domain organization of MCP_514_. MA-H1: α-helix 1 of MA subdomain; MA-H2: α-helix 2 of MA subdomain; numbers under the domains indicate the amino acid position of the domain border.

**Figure 2 microorganisms-09-01923-f002:**
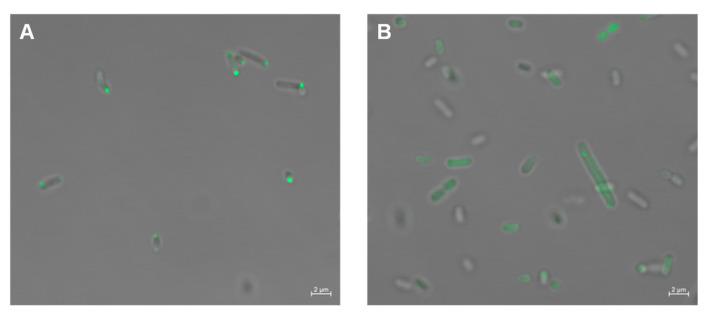
Cellular localization of MCP_514_. Plasmid expressing the eGFP–MCP_514_ fusion protein was introduced into *A. fabrum* MCP_514_-deficient mutant ∆514 (**A**) and *E. coli* strain DH5α (**B**), respectively. Bacterial cells were grown to middle-log phase and observed by using confocal laser-scanning microscopy.

**Figure 3 microorganisms-09-01923-f003:**
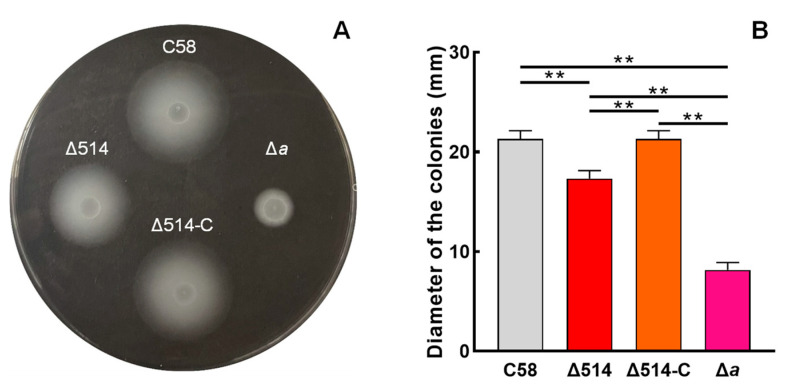
Effect of MCP_514_ deficiency on the chemotactic response of *A. fabrum*. The tested *A. fabrum* strains were grown to middle-log phase. Bacterial cells were collected and resuspended to the same OD_600nm_ (0.6). Equal amounts of cells from these cell suspensions were inoculated on the swim plates. The plates were incubated at 28 °C for 2 days. (**A**) Typical colonies of these tested *A. fabrum* strains. (**B**) The swim-ring diameters of these tested *A. fabrum* strains on the swim plate. The data represent the means ± SDs from five independent experiments in triplicate. The bars paired with two asterisks “**” represent that they are statistically different at *p* < 0.01 via the one-way ANOVA, followed by Tukey test. C58, *A. fabrum* wildtype C58; Δ514, MCP_514_-deficient mutant; Δ514-C, Δ514 mutant complemented with MCP_514_ through plasmid; Δ*a*, CheA-deficient mutant (a control of chemotaxis deficiency).

**Figure 4 microorganisms-09-01923-f004:**
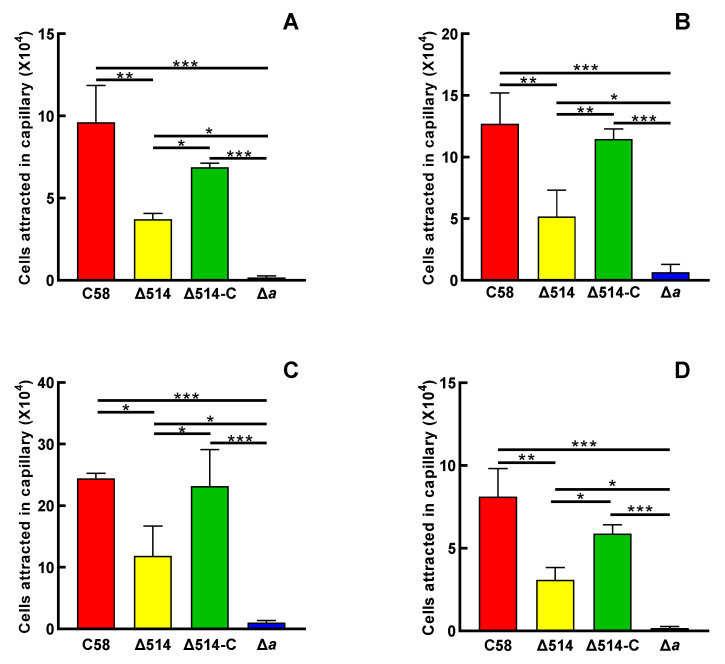
Effect of MCP_514_ deficiency on the chemotactic responses of *A. fabrum* to sucrose (**A**), valine (**B**), citric acid (**C**) and acetosyringone (**D**). *A. fabrum* cells in the middle-log phase were collected, washed and then adjusted to an OD_600nm_ of 0.1 with chemotaxis buffer. Capillary tubes containing chemotaxis buffer with 10^−6^ mol/L sucrose, 10^−3^ mol/L valine, 10^−3^ mol/L citric acid, or 10^−7^ mol/L acetosyringone were inserted into the cell suspensions for 1 h at room temperature (25 °C). Cells migrating to the capillary tube were counted by using colony count. The attracted cells are equal to the cells in the capillary tube with attractant minus the cells in the capillary tube without attractant. The data represent the means ± SDs from three independent experiments with triplicate. The bars paired with “*”, “**” and “***” marks represent that they are different in a statistical manner at *p* < 0.05, 0.01 and 0.001, respectively, via the one-way ANOVA, followed by Tukey test. The strain names in the horizontal axis are the same as in Figure 3.

**Figure 5 microorganisms-09-01923-f005:**
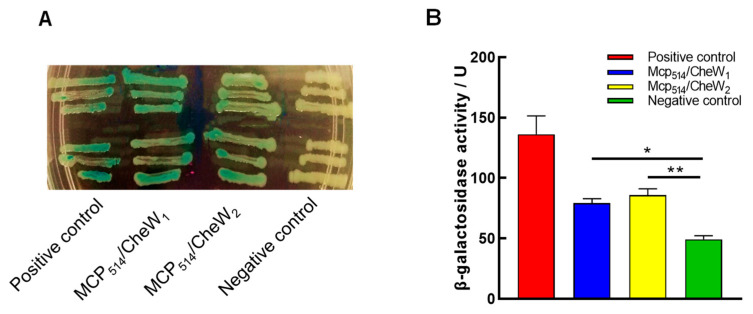
The interactions of MCP_514_ with two CheWs of *A. fabrum*. The interaction between MCP_514_ and CheW was tested by using a bacterial two-hybrid assay. All the operations followed the manual. The interaction between two tested proteins will induce the expression of β-galactosidase, and bacterial colonies will show blue color on the X-gal indicator plate. (**A**) Colony color on the X-gal indicator plate. (**B**) The activity of β-galactosidase reporter induced by the interacting proteins. Data are the means ± SDs from three independent experiments with triplicate. The bars paired with “*” and “**” marks represent that they are different in a statistical manner at *p* < 0.05 and 0.01, respectively via the one-way ANOVA, followed by Tukey test. Positive control, interaction between two known proteins (LGF2 and Gal11^p^) provided by the manufacturer; MCP_514_/CheW_1_, interaction between MCP_514_ and CheW_1_; MCP_514_/CheW_2_, interaction between MCP_514_ and CheW_2_; negative control, without any interacting proteins.

**Figure 6 microorganisms-09-01923-f006:**
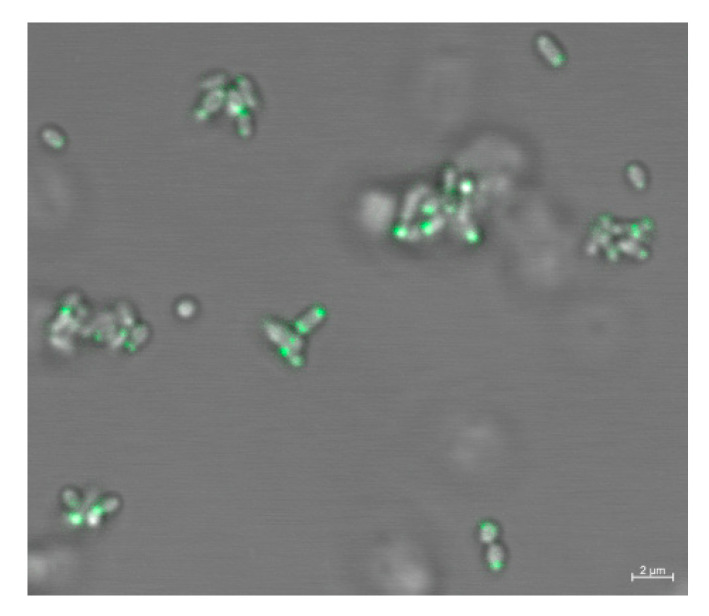
Cellular localization of MCP_514_ in CheW-deficient *A. fabrum* mutant. Plasmid expressing eGFP-MCP_514_ fusion protein was introduced into the *atu051–cheW_1_–cheW_2_* triple-deletion mutant Δ514Δ*w*. Cells were grown to the middle-log phase and observed by using confocal laser-scanning microscopy.

**Figure 7 microorganisms-09-01923-f007:**
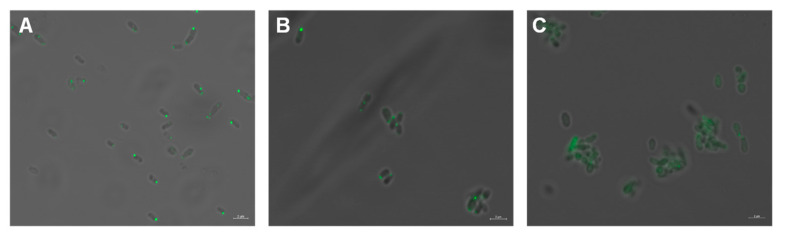
Effects of the substitution of key residue Phe328 on the cellular localization of MCP_514_. Key residue Phe328 in the hairpin subdomain of MCP_514_ was replaced by Ala, Pro or Trp to generate three MCP_514_ variants, MCP_514_^F328A^ (**A**), MCP_514_^F328W^ (**B**) and MCP_514_^F328P^ (**C**). Three MCP_514_ variants were fused to eGFP and expressed as the eGFP-MCP_514_ variant in the MCP_514_-deficient mutant ∆514, respectively. *A. fabrum* mutant ∆514 cells expressing these eGFP-MCP_514_ variant fusion proteins were grown to the middle-log phase and observed by using confocal laser-scanning microscopy.

**Figure 8 microorganisms-09-01923-f008:**
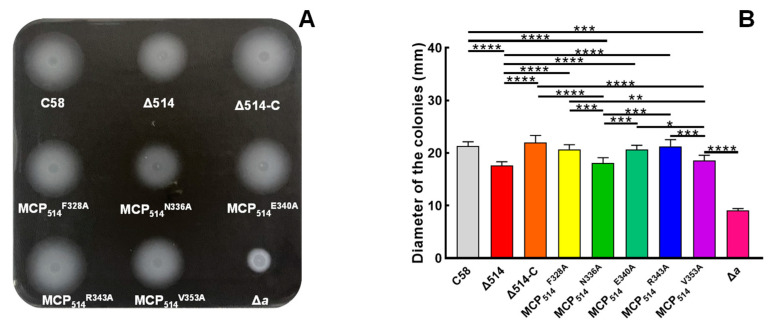
Effects of MCP_514_ variants on the chemotactic response of *A. fabrum*. The test procedure was the same as described in Figure 3. (**A**) Typical colonies of these tested *A. fabrum* strains. (**B**) The swim-ring diameters of these tested *A. fabrum* strains on the swim plate. The data represent the means ± SDs from five independent experiments in triplicate. The bars paired with “*”, “**”, “***” and “****” marks represent that they are different in a statistical manner at *p* < 0.05, 0.01, 0.001 and 0.0001, respectively, via the one-way ANOVA, followed by Tukey test. C58, *A. fabrum* wildtype C58 strain; Δ514, MCP_514_ deficient mutant; Δ514-C, Δ514 mutant complemented with native MCP_514_; MCP_514_^F328A^, MCP_514_^N336A^, MCP_514_^E340A^, MCP_514_^R343A^ and MCP_514_^V353A^ represent Δ514 mutant complemented with the corresponding single residue-substituted MCP_514_ variants, respectively; Δ*a*, CheA deficient mutant.

## Data Availability

The data that support the findings of this study are available from the corresponding author upon reasonable request.

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
