# Peer review of "The Only Chemoreceptor Encoded by che Operon Affects the Chemotactic Response of Agrobacterium to Various Chemoeffectors"

_microorganisms, 2021, doi:10.3390/microorganisms9091923_

Round 1

Reviewer 1 Report

An additional round of English language proofing for proper use of articles and plurals would be beneficial.

The manuscript presents robust and diverse data to support the story. I have a few experimental concerns that I believe should be addressed. Is the N-terminally gfp tagged MCP514 functional? This should minimally be tested by genetic complementation of motility on swim agar. 

While I think it would also be beneficial to test both N and C-terminally gfp tagged variants for their localization, N-terminal tagging plus genetic complementation would be adequate.

Similarly the F328P mutant should be tested for its affects on motility not just localization.

Section 3.5 I would argue that this is not direct evidence for effects on signal transduction specifically. That would be better assessed by the two hybrid or cell localization assays 

Lastly, please add a diagram of the che operon to figure 1

Author Response

First of all, we thank reviewer for your constructive comments and suggestions that help us to improve our manuscript. Our responses to your comments and suggestions as following:

Comment 1

I have a few experimental concerns that I believe should be addressed. Is the N-terminally gfp tagged MCP514 functional? This should minimally be tested by genetic complementation of motility on swim agar.

While I think it would also be beneficial to test both N and C-terminally gfp tagged variants for their localization, N-terminal tagging plus genetic complementation would be adequate.

Response 1

Yes, it is the main concern whether the fusion of GFP affects MCP514 function and which terminus to be fused is better for MCP514 function. Qingxuan Zhou, one author of this manuscript, once conducted this experiment in her thesis for master degree. Her results showed that C-terminally gfp tagged MCP514 variant can not restore the chemotactic function of MCP514-deficient mutant to the level of wild-type, but N-terminally gfp tagged MCP514 variant can. Therefore, we used N-terminally gfp tagged MCP514 to further investigate the function of MCP514 in this manuscript. Because her results were presented in her thesis that was collected in the thesis database, we did not present these results in this manuscript.

Comment 2

Similarly the F328P mutant should be tested for its affects on motility not just localization.

Response 2

According to previous studies on E. coli MCPs, co-localization of MCP with the other main chemotactic components and formation of MCP-CheW-CheA ternary complex are prerequisite for the chemotactic function of MCP. F328P mutant lost its polar localization. We believe that F328P mutant will definitely lose the chemotactic function. In addition, the plate used in Figure 8A is not big enough to inoculate ten strains in the same plate so that all tested strains are comparable in the same plate. Therefore, the F328P mutant was not used in this experiment.

Comment 3

Section 3.5   I would argue that this is not direct evidence for effects on signal transduction specifically. That would be better assessed by the two hybrid or cell localization assays

Response 3

This argument is reasonable. We have re-written this section. Please see the revised manuscript.

Comment 4

Lastly, please add a diagram of the che operon to figure 1

Response 4

A diagram of the che operon was added to figure 1.

Reviewer 2 Report

Methods:

Section 2.3:

-Please elaborate how atu0514, cheW1 and cheW2 were inactivated. Was it a "clean" deletion, insertion or marker exchange? Please elaborate more about the procedure and provide a physical map of the manipulated areas and PCR validation in the supplementary.

-the authors mentioned that atu0514 is part of the che operon. Have the authors checked if disruption of atu0514 had a polar effect on the rest of the operon?

Results:

*All figure legends: the authors should clarity state for each experiment how many times it was repeated: how many internal biological repeat were conducted in each experiment and how many experimental repeats.

*Figures 3, 4, 5 and 8: what was the post hoc test that was used after the ANOVA analysis? Please elaborate in the text.

3.1 A small paragraph describing the MCPs in Agrobacterium + a cartoon figure showing the physical map of the che operon (marking  atu0514 in it) will be very informative.

Line 178:  Based on what grounds the authors decided "MCP514 is a cytoplasmic chemoreceptor"? Is it the lack of transmembrane domain? Please elaborate in the text.

Line 193: Please change "text" to "test"

3.2: (195-200)/Figure 3: The experiment described in the text/figure 3 is examining swimming motility and not chemotactic response. Please correct this information in the text.  In addition, please describe the use of the cheA mutant (positive control) in the text as well as in the figure. The readers should not encounter information in the figure/figure legend that is not supported by the main text.

Line 243: please rephrase "are darker than the negative control".

Figure 5: what was the positive control that was used for this experiment (the manufacture must have specified it)?

3.5: The authors claimed that localization of atu0514 is dependent on protein-protein interaction with other MCPs but do not provide any solid experimental evidence to support this claim. Point mutations might affect protein stability and structure. Mutating these residues and demonstrating an effect on motility is not a stand alone experiment that proves this claim. To do so, the authors should demonstrated protein-protein interaction with another MCP and show that this interaction is abolished in one of the mutants that can not complement swimming  motility and are no longer be localized at the pole (F328P for instance).

Figure 8: why the F328P mutant was not used in this experiment?

Discussion:

Lines 334-338: grammar is improper. Please rephrase.

Lines 356-361: The authors did not monitor the transcriptional expression pattern of the MCPs and therefore cannot make these claimsv

Author Response

First of all, we thank reviewer for your constructive comments and suggestions that help us to improve our manuscript. Our responses to your comments and suggestions as following:

Comment 1

-Please elaborate how atu0514, cheW1 and cheW2 were inactivated. Was it a "clean" deletion, insertion or marker exchange? Please elaborate more about the procedure and provide a physical map of the manipulated areas and PCR validation in the supplementary.

Response 1

All mutants are “clean” deletion mutants. Method for constructing mutant was re-written in the revised manuscript. The procedure, physical map, PCR screening and sequencing validation were provided in the supplementary. Please see the revised supplementary.

Comment 2

-the authors mentioned that atu0514 is part of the che operon. Have the authors checked if disruption of atu0514 had a polar effect on the rest of the operon?

Response 2

The evidence that the introduction of MCP514-expressing plasmid can fully restore the chemotactic response of the MCP514-deficient mutant to the level of the wild-type, demonstrates that the deletion of atu0514 did not have a polar effect on the rest of the operon.

Comment 3

*All figure legends: the authors should clarity state for each experiment how many times it was repeated: how many internal biological repeat were conducted in each experiment and how many experimental repeats.

*Figures 3, 4, 5 and 8: what was the post hoc test that was used after the ANOVA analysis? Please elaborate in the text.

Response 3

All the mentioned questions were elaborated in all figure legends and the text. Please see the revised manuscript.

Comment 4

3.1 A small paragraph describing the MCPs in Agrobacterium + a cartoon figure showing the physical map of the che operon (marking  atu0514 in it) will be very informative.

Response 4

A diagram of the che operon was added to figure 1. A brief description of the che operon was added to the revised manuscript. Please see the revised manuscript.

Comment 5

Line 178:  Based on what grounds the authors decided "MCP514 is a cytoplasmic chemoreceptor"? Is it the lack of transmembrane domain? Please elaborate in the text.

Response 5

Yes, based on the the lack of transmembrane domain. This was elaborated in the revised manuscript.

Comment 6

Line 193: Please change "text" to "test"

Response 6

Corrected. Please see the revised manuscript.

Comment 7

3.2: (195-200)/Figure 3: The experiment described in the text/figure 3 is examining swimming motility and not chemotactic response. Please correct this information in the text.  In addition, please describe the use of the cheA mutant (positive control) in the text as well as in the figure. The readers should not encounter information in the figure/figure legend that is not supported by the main text.

Response 7

The required information was added to section 3.2 and figure 3. Please see the revised manuscript.

Comment 8

Line 243: please rephrase "are darker than the negative control".

Response 8

Corrected. Please see the revised manuscript.

Comment 9

Figure 5: what was the positive control that was used for this experiment (the manufacture must have specified it)?

Response 9

The proteins of the positive control were added to the legend of figure 5.

Comment 10

3.5: The authors claimed that localization of atu0514 is dependent on protein-protein interaction with other MCPs but do not provide any solid experimental evidence to support this claim. Point mutations might affect protein stability and structure. Mutating these residues and demonstrating an effect on motility is not a stand alone experiment that proves this claim. To do so, the authors should demonstrated protein-protein interaction with another MCP and show that this interaction is abolished in one of the mutants that can not complement swimming  motility and are no longer be localized at the pole (F328P for instance).

Response 10

This argument is reasonable. We have re-written this section and revised some sentences in the discussion section. Please see the revised manuscript.

Comment 11

Figure 8: why the F328P mutant was not used in this experiment?

Response 11

According to previous studies on E. coli MCPs, co-localization of MCP with the other main chemotactic components and formation of MCP-CheW-CheA ternary complex are prerequisite for the chemotactic function of MCP. F328P mutant lost its polar localization. We believe that F328P mutant will definitely lose the chemotactic function. In addition, the plate used in Figure 8A is not big enough to inoculate ten strains in the same plate so that all tested strains are comparable in the same plate. Therefore, the F328P mutant was not used in this experiment.

Comment 12

Lines 334-338: grammar is improper. Please rephrase.

Response 12

Rephrased. Please see the revised manuscript.

Comment 13

Lines 356-361: The authors did not monitor the transcriptional expression pattern of the MCPs and therefore cannot make these claimsv

Response 13

These sentences were deleted. Please see the revised manuscript.

Round 2

Reviewer 1 Report

Line 82 chemotaxis

Response 1

If the GFP tagged version has been demonstrated as functional discuss that in the manuscript and add the appropriate citation to the thesis.

Response 2

The motility assay is not difficult to perform nor time intensive. The plate size limit is not a good argument for not performing this experiment. 

Please complete this test with an appropriate subset of strains that fit on your testing plates.

Author Response

Thank you for your review again.

Ref. to "Line 82 chemotaxis"

Response: Corrected

Ref. to "If the GFP tagged version has been demonstrated as functional discuss that in the manuscript and add the appropriate citation to the thesis."

Response: The thesis was cited in the revised manuscript and one sentence was added to the text.

Ref. to "The motility assay is not difficult to perform nor time intensive. The plate size limit is not a good argument for not performing this experiment.

Please complete this test with an appropriate subset of strains that fit on your testing plates."

Response: Yes, the absence of the F328P mutant in figure 8 is a little bit imperfect and performing the motility assay is not difficult. However, since early August Yangzhou city has been in the hard lockdown due to the COVID-19 pandemic. Campuses and labs in the university are closed. We all stay home and work online. Hopefully, we can get into the lab after two or three weeks. Consequently, if we must provide the mentioned experimental result, we need three or more weeks to revise the manuscript.

Reviewer 2 Report

The authors properly addressed all the concerns that arose during the first review cycle.

The quality and the presentation of the work are good and the manuscript is well written.

The authors should double check the manuscript for minor typos/spelling errors.

 For instance:

Line 134: "duoble-deletion" should ne "double-deletion"

Good luck

Author Response

Thank you for your review again.

Some minor spelling errors (including the mentioned error) were checked and corrected.